# Navigating the Complexities of Traumatic Encephalopathy Syndrome (TES): Current State and Future Challenges

**DOI:** 10.3390/biomedicines11123158

**Published:** 2023-11-27

**Authors:** Arman Fesharaki-Zadeh

**Affiliations:** Department of Neurology and Psychiatry, Yale University School of Medicine, New Haven, CT 06510, USA; arman.fesharaki@yale.edu

**Keywords:** chronic traumatic encephalopathy (CTE), Alzheimer’s disease (AD), traumatic encephalopathy syndrome (TES), traumatic brain injury (TBI), repetitive head impacts (RHI)

## Abstract

Chronic traumatic encephalopathy (CTE) is a unique neurodegenerative disease that is associated with repetitive head impacts (RHI) in both civilian and military settings. In 2014, the research criteria for the clinical manifestation of CTE, traumatic encephalopathy syndrome (TES), were proposed to improve the clinical identification and understanding of the complex neuropathological phenomena underlying CTE. This review provides a comprehensive overview of the current understanding of the neuropathological and clinical features of CTE, proposed biomarkers of traumatic brain injury (TBI) in both research and clinical settings, and a range of treatments based on previous preclinical and clinical research studies. Due to the heterogeneity of TBI, there is no universally agreed-upon serum, CSF, or neuroimaging marker for its diagnosis. However, as our understanding of this complex disease continues to evolve, it is likely that there will be more robust, early diagnostic methods and effective clinical treatments. This is especially important given the increasing evidence of a correlation between TBI and neurodegenerative conditions, such as Alzheimer’s disease and CTE. As public awareness of these conditions grows, it is imperative to prioritize both basic and clinical research, as well as the implementation of necessary safe and preventative measures.

## 1. Background

Chronic traumatic encephalopathy (CTE) is a distinct neurodegenerative disease and is often associated with a history of repetitive head impacts (RHI) in the context of sports or combat settings. The defining neuropathological characteristic of CTE includes hyperphosphorylated tau at the depths of cortical sulci and peri-vascular regions [1]. CTE was first reported in a group of boxers, who were described as “punch drunk” by Martland in 1928 [2]. The report described a group of boxers who had suffered repetitive head blows throughout their sporting careers, with clinical presentations of behavioral symptoms as well as severe memory and neurocognitive deficits. The newly defined neurodegenerative condition was labeled “dementia pugilistica” [3], and eventually CTE in 1949 [4]. More recently, Omalu et al. reported finding evidence of CTE in three retired football players [5,6,7]. McKee reported similar findings in three new individuals when reviewing the world literature on CTE, including one football player, as well as a multitude of reports and case studies of evidence of CTE in athletes and veterans who were exposed to repetitive head trauma.

In 2014, research diagnostic criteria for traumatic encephalopathy syndrome (TES) were proposed for use in clinical research settings to diagnose CTE in patients while alive by Montenigro et al. [8]. They proposed five general criteria for diagnosis of TES, which included (1) a history of multiple head impacts, (2) no other neurological disorders accounting for all the clinical features, (3) clinical features present for a minimum of 12 months, (4) at least one core clinical feature must be present, and (5) the presence of at least two supportive features.

The core features included cognitive deficits, behavioral symptoms, and mood symptoms. The cognitive symptoms, supported by detailed neuropsychological assessments, included changes in episodic memory, executive function, and attention, as defined by 1.5 standard deviations below normal. Behavioral symptoms included verbal or physical aggression, while mood symptoms included feeling depressed or hopeless. The supportive features included impulsivity, anxiety, apathy, paranoia, suicidality, headache, motor signs including dysarthria and dysgraphia, documented decline, as well as a delayed onset.

Mez et al. examined the validity of 336 brain donors with a prior history of repetitive head impacts due to etiologies, including contact sports, military settings, and/or due to violent trauma [9]. The TES criterion had a reported sensitivity and specificity of 0.97 and 0.21, respectively. Cognitive symptoms were significantly associated with CTE pathology. Modifying the TES criterion using cognitive deficits resulted in improved specificity (0.48) and a mild reduction in sensitivity (0.90), respectively. The authors found that having cognitive symptoms was significantly associated with CTE pathology, increasing the odds by 3.6-fold.

This review endeavors to offer a comprehensive exploration of the underlying pathophysiological mechanisms of traumatic encephalopathy syndrome (TES) and chronic traumatic encephalopathy (CTE). Additionally, it provides an in-depth survey of the present neuroimaging and plasma biomarkers employed in the diagnosis of traumatic brain injury (TBI). Furthermore, it scrutinizes the currently utilized clinical regimens in TBI treatment (refer to Figure 1). Given the rapid evolution of the research field concerning TES, CTE, and TBI at large, this review aims to encapsulate some of the most promising research trajectories in the field [10].

## 2. Neuropathology

Coresellis et al., 1973 assessed 15 boxers who were diagnosed with dementia pugilistica. The reported neuropathological findings included neurofibrillary tangles (NFTs) without accompanying amyloid plaques, more prominently found in the medial temporal lobes and brainstem, substantial nigral atrophy with NFTs, cerebellar tonsils gliosis, and cavum septum pellucidum [11]. The neuropathological characterization of CTE has undergone further iterations [12]. As per the National Institute of Neurological Disorders and Stroke (NINDS)-funded study entitled “Understanding Neurologic Injury in Traumatic Encephalopathy” (UNITE), CTE’s characteristic neuropathological changes include NFTs in astrocytes, accumulated around blood vessels, and at the depth of sulci in an irregular fashion. As per NINDS panel consensus, the pattern of p-tau is distinct from other neurodegenerative conditions. Furthermore, the tau filament in CTE has been shown to have a unique conformation of the β-helix region with a hydrophobic cavity [13,14].

McKee at al. reported ß-amyloid deposition, an essential feature of AD, in 43% of CTE cases [15]. Gardner et al. examined the autopsies of 85 athletes and found that only 20% had pure CTE, 52% with CTE and another co-morbid neuropathology, 5% with no CTE, and 24% with no observed neuropathology. These studies, in turn, highlight the inherent heterogeneity of CTE [16].

As defined by McKee et al., CTE pathological progression could be conceptualized in 4 distinct stages. In stage 1, the brain has typically normal weight, with NFTs and perivascular p-tau and astrocytic tangles predominantly in the superior and dorsolateral frontal cortices. In stage 2, there are more NFTs throughout the superficial cortical layers, as well as locus coeruleus and substantia innominata. In stage 3, there is a reduction in brain weight with accompanying cortical atrophy and ventricular dilation, with frequent cavum septum pellucidum. In addition, there is notable depigmentation of locus coeruleus and substantia nigra, as well as atrophic changes of mamillary bodies, thalamus, and hypothalamus, as well as white matter tracts, including corpus callosum. There are also widespread NFTs in various subcortical areas, including olfactory bulbs, amygdala, hippocampi, hypothalamus, mamillary bodies, nucleus basalis of Meynert, substantia nigra, dorsal and median raphe nuclei, locus coeruleus, and entorhinal cortex. In stage 4, there is substantial global cortical atrophy, including medial temporal lobes, and thalamus, hypothalamus, and mamillary bodies, with complete depigmentation of the locus coeruleus and substantia nigra [1] (Figure 2).

CTE staging criteria are based on a series of multiple case studies and are largely cross-sectional and not longitudinal [1]. The study of Bieniek et al. reported CTE in 32% of the 66 studied athletes [18]. Another study by Stern et al. focused on the clinical presentation of 36 neuropathologically confirmed CTE patients [19]. The study reported two distinct clinical phenotypes: A younger (n = 22) group initially presented with behavioral and mood changes, and an older (n = 11) group presented predominantly with cognitive changes. Another notable study involving retired NFL players by Hampshire et al. reported abnormal connectivity changes on functional MRI (fMRI) involving the dorsal frontoparietal network [20].

## 3. Potential TES Biomarkers

As CTE is largely a postmortem diagnosis, the precise diagnosis of traumatic encephalopathy syndrome remains elusive. The clinical diagnosis of TES largely depends on neuroimaging, along with CSF and plasma biomarkers (see Table 1), that are measurable in a clinical setting. Although there is no consensus on a set of TES biomarkers, there is agreement on a set of converging neuroimaging; therefore, CSF/plasma markers warrant further discussions.

### 3.1. Neuroimaging

Asken et al. studied the structural MRI scans of nine patients with TES [37]. The regions of interest (ROIs) included the dorsal frontal, ventral frontal, temporal, parietal, occipital, thalamus, and medial temporal lobes. All nine patients had reported cavum septum pellucidum (CSP), with no clear differences between patients with high CTE vs. those without CTE. There was prominent medial temporal atrophy amongst all nine patients. Other ROIs reported to have undergone notable atrophy included the thalamus, ventral frontal cortex (8/9), dorsal frontal cortex (8/9), and orbitofrontal cortex, as well as the right posterolateral frontal cortex [38].

The use of diffusion tensor imaging (DTI) has not been standardized for the diagnosis and prognosis of TES patients. There are a number of promising studies, including Strain et al., based on the DTI imaging analysis of 26 retired NFL players, which reported a significant association between depression and integrity of white matter [38]. These patients also had an increase in deep white matter lesions on T2-weighted fluid inversion recovery (FLAIR), compared with matched unimpaired NFL player control subjects [37]. In the study by Asken et al. [38], six patients underwent antemortem DTI (n = 3 with High CTE; n = 1 with Low CTE; n = 2 with no CTE). All six patients had significantly diminished fractional anisotropy (FA) along the fornix, irrespective of neuropathological diagnoses. Five of the six patients had significantly decreased FA in the genu of the corpus callosum (genu CC ROI median W score = −1.24) and medial temporal white matter in the areas of the uncinate fasciculus and cingulum–hippocampal bundle.

In another study by Asken et al. [37], five patients (n = 2 High CTE, n = 1 Low CTE, n = 2 no CTE) underwent FDG-PET imaging. Major hypometabolic ROIs included the thalamus (4/5 patients), medial temporal lobes (4/5 patients), and left dorsal frontal cortex (3/5 patients). Another promising PET imaging modality includes the use of tau-based ligands, which have already been utilized in AD patients [39]. The exploration of the use of tau PET imaging for the identification of p-tau aggregates that would be potentially specific and sensitive for CTE patients is currently in progress.

In a study by Stern et al. [40] involving 26 former NFL players and 31 control subjects, flortaucipir positron-emission tomography (PET) and florbetapir PET were employed to measure the deposition of tau and amyloid-beta, respectively, in the brains of former NFL players with cognitive and neuropsychiatric symptoms, and in asymptomatic men with no history of traumatic brain injury. The regional tau standardized uptake value ratio (SUVR), which is the ratio of radioactivity in a cerebral region to that in the reference cerebellum, was used to explore the associations of SUVR with symptom severity and with years of football play in the former-player group versus the control group. The inclusion criteria for the former players were male sex, age 40 to 69 years, a minimum of 2 years playing football in the NFL, a minimum of 12 years of total tackle football experience, and cognitive, behavioral, and mood symptoms reported by the participant through telephone screening. Each participant underwent flortaucipir PET, florbetapir, and T1-weighted volumetric MRI of the head. The mean flortaucipir SUVR was higher among former players than among controls in three regions of the brain: bilateral superior frontal (1.09 vs. 0.98; adjusted mean difference, 0.13; 95% confidence interval [41], 0.06 to 0.20; *p* < 0.001); bilateral medial temporal (1.23 vs. 1.12; adjusted mean difference, 0.13; 95% CI, 0.05 to 0.21; *p* < 0.001); and left parietal (1.12 vs. 1.01; adjusted mean difference, 0.12; 95% CI, 0.05 to 0.20; *p* = 0.002). All of the former NFL players in the study reported cognitive symptoms, and more than 35% had impaired delayed recall scores on an objective memory test.

### 3.2. CSF

CSF biomarkers, including neurofilament light (NfL) and tau protein, are potential CTE biomarkers, as they have been shown to be elevated in boxers within days to weeks following injury [37]. A prior study reported total tau elevation (>3.56 pg/mL) in former NFL players, including cases of remote injuries. However, the total tau levels did not correlate with neurocognitive deficits [42]. The t-tau elevation has also been reported in other neurodegenerative conditions, such as AD and FTD, as well as cerebrovascular conditions, limiting its utility [42], but it has less specificity for CTE.

### 3.3. Plasma

Asken et al. examined antemortem plasma GFAP, NfL, and total tau for eight of the nine patients with TES [38]. Five patients had longitudinal GFAP and NfL data, and two had longitudinal total tau data. Most patients had elevated plasma GFAP, NfL, and total tau at their initial visit compared to age-matched healthy controls. Three of five patients with longitudinal GFAP and NfL data demonstrated increasing concentrations over time, and four had increasing NfL over time.

In aging cohorts, plasma GFAP was tightly linked to AD-related amyloid ß42 plaque [21,43,44]. The mechanisms underlying plasma GFAP changes in patients without AD remain to be determined but could reflect astrocytic dysfunction and inflammation in non-AD disease pathogenesis. AD-based biomarkers such as Aß-PET, CSF amyloid ß42 and phosphorylated tau, or plasma phosphorylated tau in studies of patients with previous repetitive head impacts and TES may add to more specific biomarker signatures that are specific to CTE and minimize the risk of misattributing biomarker changes to AD (co)pathology.

Neuron-specific enolase (NSE), also known as gamma-enolase or enolase 2, exists in mature neurons and neuroendocrine cells [20]. NSE has been reported to be elevated in the blood of both mTBI, as well as more severe TBI patients [23,24,45,46,47,48]. NSE is also abundantly expressed in red blood cells, making it less specific and requiring hemolysis correction for accurate blood measurement [49]. Ubiquitin C-terminal hydrolase-L1 (UCH-L1) is a protein that mainly resides in the neuronal cell body cytoplasm [32,46,50,51,52,53]. UCH-L1 was first reported to be released into CSF and serum among severe TBI patients. The use of CSF UCH-L1 is a potentially robust clinical outcome predicting marker of mortality following non-penetrating TBI [54]. UCHL-1 was also reported to be elevated in serum/plasma in mTBI, including athletes after concussion [30,55]. Based on the results of a multi-center TBI study (ALERT-TBI), GFAP and UCHL-1 have shown high sensitivity (0.976) and negative predictive value (0.996) for the detection of traumatic intracranial injury in the acute setting [56]. These robust TBI biomarker findings ultimately led to FDA approval of GFAP and UCHL-1 to aid in TBI evaluation in a clinical setting [57].

S100B, an astroglial calcium-binding protein, has been extensively studied as a TBI marker [22,26] and of various degrees of injury severity [27,58]. An important confound is the multiple potential sources of S100B, which includes adipose tissues, as well as cardiac/skeletal muscles. S100B has been reported to be elevated in orthopedic trauma without accompanying head injuries [59]. However, S100B remains a sensitive predictive marker for CT abnormality and the development of post-concussive syndrome (PCS) among mTBI patients [60,61,62].

C-terminal breakdown products (BDPs), which include SBDP120, SBDP 145, and SBDP150, are byproducts of calpain and caspase-3. These markers have been shown to be elevated in preclinical models of TBI, as well as human CSF samples [28,59,63,64,65,66,67,68,69]. Another spectrin product, the *N*-terminal spectrin fragment (SNTF), was also reported to be elevated after concussion. One potential confound is the fact that the αII-spectrin protein is expressed in other organs and peripheral blood mononuclear cells (PBMC), limiting its clinical interpretation [28,59,63,69].

Myeline basic protein (MBP) is an oligodentrocyte-based protein and an essential component of the myelin sheath. TBI leads to the activation of calpain and an increase in MBP degradation. MPB has been reported to be elevated in severe pediatric and adult TBI patients [54,70].

A novel set of TBI biomarkers include dendritic protein microtubule-associated protein-2 (MAP-2) [31,51], brain-derived nerve growth factor (BDNF) [29], and postsynaptic protein neurogranin [71]. Moreover, microRNA (miRNA) has been reported to be elevated in biofluid (CSF, serum, or plasma) in a number of preclinical TBI models [33]. There are also a number of candidate miRNA biomarkers reported in clinical TBI in both mild and severe cases [20,34,72,73].

There are reported CSF levels of MV/E containing SBDPs, synaptophysin, UCHL-1, and GFAP [74,75]. Elevated levels of circulating tau-containing exosomes has been proposed to be a promising predictive risk marker for CTE in chronic TBI patients [35].

TBI induces a cascade of secondary biological phenomena, which includes the production of a series of pro- and anti-inflammatory cytokines [36]. In individuals with severe TBI, increased levels of (IL)-6, IL-1, IL-8, IL-10, and tumor necrosis factor-alpha (TNFα) were associated with worse clinical outcomes [76]. Based on a recent meta-analysis, IL-6 was shown to have robust potential as a pro-inflammatory marker in acute mild TBI patients [77]. Il-6 has also been reported to have potential prognostic biomarker value for clinical outcomes post-TBI [78]. Moreover, as one of the outcomes of the Transforming Research and Clinical Knowledge in Traumatic Brain Injury (TRACK-TBI) study, high-sensitivity C-reactive protein (hsCRP) measured within 2 weeks of TBI was found to be a prognostic biomarker of disability 6 months later [79]. Intriguingly, post-traumatic stress disorder (PSTD), a highly co-morbid psychiatric illness in brain injury patients [80], has been shown to be a pro-inflammatory condition associated with the elevation of pro-inflammatory markers, including CRP, IL-6, and TNFα [81]. Depression, another highly co-morbid psychiatric condition with TBI [82], has also been shown to be associated with low-grade inflammation, as manifested by mildly elevated CRP levels [83].

## 4. Treatment

Although there are no established treatments for TES, there are pharmacological and non-pharmacological treatment options for the treatment of TBI/TES symptoms. The following section provides a wide scope of potential pharmacological and non-pharmacological treatment options for TBI/TES patients.

### 4.1. Non-Pharmacological Management

Potential non-pharmacological TBI regimens include outpatient regular cognitive rehabilitation therapy, mood and psychotherapy-focused treatment, including regimented cognitive behavioral therapy and mindfulness/stress reduction techniques, as well as a Mediterranean diet and aerobic exercise. Other therapeutic modalities include vestibular rehabilitative therapy, occupational/ocular therapy, and physical/motor therapy sessions when indicated [84].

### 4.2. Clinical Pharmacological Management

Currently, there are no FDA-approved disease-modifying regimens available for chronic traumatic encephalopathy (CTE). The existing treatments are considered “off-label” and primarily focus on alleviating symptoms. To address memory impairment, which is a common issue in CTE, medications originally developed for Alzheimer’s disease, such as galantamine, donepezil, and rivastigmine, have been repurposed for CTE patients [84]. Additionally, to tackle apathy symptoms, stimulants like methylphenidate and dopamine agonists such as carbidopa/levodopa, pramipexole, amantadine, and memantine may be employed. Amantadine has been the subject of extensive examination in moderate to severe TBI patients. A recent meta-analysis, encompassing 14 clinical trials and 6 observational studies, demonstrated the cognitive benefits of amantadine for this patient population. Notably, the improvements in cognition were more prominent in younger patients with less severe TBIs [85]. Stimulants can also prove beneficial in managing attention and concentration deficits. When dealing with depression and anxiety symptoms, selective serotonin reuptake inhibitors (SSRIs) like sertraline and escitalopram can be used, but caution is advised due to the potential risk of suicidality, as suicide cases have been documented in CTE [82]. Another promising approach for addressing working memory (WM) deficits resulting from traumatic brain injury involves the use of an alpha-2-adrenergic receptor agonist known as guanfacine. Through functional MRI imaging, a study by McAllister et al. demonstrated improvements in verbal WM in 13 mild TBI patients one month after their injury. Moreover, the group treated with guanfacine exhibited increased activation in WM circuitry, particularly in the prefrontal cortex (PFC) region [86].

### 4.3. Clinical Use of Nutraceutical Regimen

There is an increased level of reactive oxygen species (ROS) and reactive nitrogen species (RNS) production post-TBI due to the excitotoxic nature of injury [87]. Anti-oxidants have been widely studied as a potent treatment modality to diminish the level of ROS and RNS post-injury [88]. There are several anti-oxidant therapies used in the treatment of TBI, including ascorbic acid (vitamin C), N-acetylcysteine (NAC), flavonoids, resveratrol, alpha-tocopherol (vitamin E), coenzyme Q10, carotenoids, omega-3 fatty acids, and Pycnogenol^®^ [87,88].

### 4.4. Preclinical Investigational Pharmacological Intervention

A wide array of preclinical models of TBI have been utilized, including fluid percussion injury (FPI), the blast wave injury model, the weight drop injury model (WDI), and controlled cortical impact models. More novel preclinical injury models include the closed-head impact model of engineered rotational acceleration (CHIMERA) and closed-head projectile concussive impact (PCI). The TBI preclinical models have induced a range of neuropathological changes, including microgliosis, tauopathy, endoplasmic reticulum stress (ER), excitotoxicity, and white matter injury [89,90]. These models also capture the long-lasting, chronic cognitive deficits and mood fluctuations associated with TBI [90,91,92,93,94,95,96,97]. Moreover, there are several prospective treatment targets, encompassing tau acetylation, tau phosphorylation, and the realms of neuroinflammation and immunotherapy.

### 4.5. Targeting Tau Acetylation

Tau phosphorylation follows tau acetylation, and it is often induced by neuroinflammation and oxidative stress [84,96,98]. A tau acetylation modulating agent is salsalate, which has been shown to reduce inflammation, provide neuroprotection, and enhance neurogenesis via gene regulation. Salsalaye has been largely studied in preclinical models [99,100]. Methylene blue, which modulates K280/K281 acetylation activity, was reported to increase neuroprotection, diminish behavioral deficits and mood changes, and minimize neuronal degeneration, neuroinflammation, lesion volume, microgliosis, and mitochondrial dysfunction in TBI rodent models [101,102,103]. Histone deacetylase 6 (HDAC) and sirtuins (SIRT1 and SIRT2) increased tau deacetylation, presenting another potential treatment methodology that targets the same pathway mechanism but in a different manner [104].

### 4.6. Targeting Tau Phosphorylation

Inhibition of kinases has been studied in preclinical TBI models. One such target kinase is glycogen synthase 3 beta (GSK-3β), induced by p-tau, which leads to further downstream tau phosphorylation and amyloidapathy, diminishing anti-oxidant defenses such as nuclear factor E2-related factor 2 (Nrf2) [84,105,106,107]. Agents such as dimethyl fumarate (DMF) and lithium modulate GSK-3β activity and have been reported to reduce neurodegenerative processes, diminish lesion size in preclinical TBI models, and improve neurocognitive outcomes. Lithium may also modulate behavioral symptoms such as mood, impulsivity, and suicidal behavior [108]. Another kinase inhibitor, roscovitine, inhibits cyclin-dependent kinase (CDK) and has been reported to modulate neuroinflammation, diminish neurodegeneration, and improve cognitive outcomes in rat preclinical TBI models [108,109,110,111,112]. Another preclinical study reported a potential synergistic role in diminishing p-tau levels in repetitive mild TBI models [97]. Using a preclinical CTE model based on combined repetitive mild TBI and chronic stress, Tang and Fesharaki–Zadeh et al. examined the long-term pharmacological use of Fyn kinase inhibition, AZD0530. Post-injury Fyn inhibition led to a reduction of focal phospho-tau accumulation, as well as neurobehavioral rescue as measured by rescuing object recognition and improving spatial memory function (Figure 3) [90].

### 4.7. Targeting Inflammation

Damage and cellular demise lead to the extracellular release of various ions, molecules, and proteins collectively known as damage-associated molecular patterns (DAMPs) [113,114]. These DAMPs encompass ATP and K+, double-stranded DNA, and the high mobility group 1 (NMG1) chromatin protein. ATP binds and activates P2 × 7 receptors, while elevated K+ stimulates pannexin receptors [115]. DAMPs bind extracellular receptors that activate intracellular inflammasomes [116]. Activated inflammasomes in neurons and astrocytes convert pro-IL-1β and pro-IL-18 into their biologically active forms [117]. Extracellular IL-1β and IL-18 levels increase acutely post-injury and are the main inducers of microglia and other early inflammatory processes [115]. TNFα, IL-6, IL-12, and interferon γ are additionally released in the acute phase of injury [115]. Neurovascular changes, infiltration of peripheral inflammatory cells, and activation of resident microglia and astrocytes lead to a more global release of cytokines, chemokines, and bioactive lipids [118,119]. The alteration of microglia activation is a key event in switching from inflammation with early and largely deleterious effects to a later phase of tissue repair and remodeling [118]. Microglia can differentiate into either pro-inflammatory M1 or anti-inflammatory M2 phenotypes [120]. M1 microglia intensify inflammation, bolster the presence of pro-inflammatory cells, and facilitate the clearance of apoptotic cells. They secrete pro-inflammatory cytokines, including IL-1β, TNFα, and IL-6, along with chemokines that attract more inflammatory cells to the site of injury. Moreover, M1 microglia amplify oxidative stress through elevated expression of NADPH oxidase and iNOS [121]. The M2 microglial has been reported to have an anti-inflammatory role [122]. The precise identification of the inflammatory mediators essential for achieving optimal therapeutic effects remains an ongoing challenge [123].

Previous investigations have aimed at addressing the intricate inflammatory cascade and metabolic changes in preclinical models of CTE. A recent study specifically centered on the potential application of the pyrimidine derivative OCH, which is believed to safeguard mitochondrial function and maintain adequate ATP synthesis following traumatic brain injury (TBI) [124]. OCH demonstrated enhancements in ATP production, respiratory efficiency, and cerebral blood flow, coupled with reductions in glycolysis activity, CTE biomarker levels, and β-amyloid concentrations. Furthermore, OCH treatment effectively preserved sensorimotor function [124]. The use of salubrinal (SAL), a stress modulator, significantly diminished ER stress, oxidative stress, pro-inflammatory cytokines, and inducible nitric oxide synthase. SAL treatment also reduced impulsive-like behavior in rodent models of repetitive TBI [125]. Calpain-2 has been implicated in the progression of neurodegeneration after TBI. The application of a selective calpain-2 inhibitor, known as C2I, resulted in a significant reduction in calpain-2 activation. This intervention effectively halted the elevation of tau phosphorylation and TDP-43 alterations, curbed astrogliosis and microgliosis, and successfully mitigated cognitive impairment in a preclinical model of repeated mild traumatic brain injury [126]. Inhibiting monoacylglycerol lipase (MAGL), responsible for the metabolism of 2-arachidonoylglycerol (2-AG), yielded significant reductions in neurodegeneration, tau phosphorylation, TDP-43 aggregation, astrogliosis, and pro-inflammatory cytokines. This intervention also resulted in improved cognitive outcomes in a rodent model of repetitive mild TBI. Additionally, the application of 2-AG enhanced blood–brain barrier integrity and reduced the expression of inflammatory cytokines when utilized in a preclinical CHI rodent model [116].

Glucocorticoids exert a broad anti-inflammatory effect by inhibiting the synthesis of interleukins and bioactive lipids. They also suppress cell-mediated immunity and reduce leukocyte count and activity [117]. Despite several preclinical studies, none have investigated whether the anti-inflammatory properties of dexamethasone translate into improved brain function [126]. Clinical trials have yielded limited success, likely due to a narrow therapeutic window [127]. A significant phase III trial, known as CRASH (corticosteroid randomization after significant head injury), included 10,008 adults with TBI and a Glasgow Coma Score (GCS) ≤ 14 [127]. Within 8 h of the injury, these patients received a 48-h infusion of methylprednisolone or a placebo. Intriguingly, the methylprednisolone group exhibited a higher risk of mortality compared to the placebo group, irrespective of the injury severity, thus diminishing the potential clinical efficacy of this regimen.

Non-steroidal anti-inflammatory drugs (NSAIDs) represent a class of medications known for their potent analgesic, antipyretic, and anti-inflammatory properties achieved through the inhibition of COX-1 and COX-2 [128]. COX-2 selective drugs such as carprofen, celecoxib, meloxicam, nimesulide, and rofecoxib have undergone testing in various preclinical TBI models [129]. Despite their anti-inflammatory potential, these agents have not proven sufficiently effective in targeting COX-1 or COX-2, making them less promising as therapeutic options for TBI treatment [114]. TNFα, a pro-inflammatory cytokine induced post-TBI, was targeted using HU-211, a synthetic cannabinoid, leading to sustained improvements in various cognitive and motor functions when administered within two hours post-injury [118]. Another TNFα antagonist showed effectiveness in reducing IL-1β and IL-6 at 3 days post-injury and TNFα at both 3 and 7 days post-injury [129]. Similarly, IL-1β, an acute pro-inflammatory cytokine post-injury, was modulated in mice overexpressing IL-1ra, resulting in reduced edema and improved neurological scores [130]. Anakinra, a human IL-1 receptor antagonist, when administered two hours after traumatic brain injury, had a limited impact in various assessments [131]. In a clinical trial, administering anakinra within 24 h of injury modified the neuroinflammatory response, but the study’s size prevented a clear determination of its therapeutic effect [132].

Rolipram, a phosphodiesterase IV inhibitor, altered both histology and function when administered 30 min before injury [114]. When given 30 min after injury, rolipram similarly lowered IL-1β and TNFα levels three hours after injury. However, the lesion size was increased compared to vehicle controls [133].

Minocycline is a lipophilic tetracycline-based antibiotic that can cross the BBB with anti-inflammatory action at higher concentrations [134]. Multiple prior studies have demonstrated the anti-inflammatory effects of minocycline [25,135]. Administering minocycline between 5 min and 1 h after injury enhanced performance in various behavioral tasks, such as novel object recognition, the elevated plus maze, the Morris water maze, and active place avoidance in preclinical TBI studies [25,136]. The reduction in IL-1β production is believed to be the mechanism behind minocycline’s inhibitory effect on microglia [137]. Combining minocycline with N-acetylcysteine (NAC) synergistically improved memory in the active place avoidance (APA) task, a complex spatial memory test [136].

Progesterone, a gonadal hormone, has various anti-inflammatory effects. When given 30 min after injury, progesterone initially raised IL-1β levels at 6 h, followed by a decrease at 24 h. It inhibited IL-6 at both 6 and 24 h post-injury, reduced TNFα at 6 h, and increased TGFβ levels at 24 h [138]. In the PROtect phase II trial, patients receiving progesterone within 11 h of injury had a lower 30-day mortality rate than those receiving a placebo. Patients with moderate traumatic brain injury showed better outcomes on clinical scales. However, a large Phase III PROtect III trial involving 882 patients was terminated as it showed no significant effect of progesterone on functional recovery compared to placebo based on the extended Glasgow Outcome Score 6 months post-injury [128]. Another trial (SYNAPSE) with 569 severe TBI patients did not find differences in outcomes between progesterone and control groups based on various assessments at different time points.

Erythropoietin, responsible for regulating the growth of red blood cell precursors in the bone marrow, has demonstrated anti-apoptotic, anti-oxidative, angiogenic, and neurotrophic effects in various preclinical models of traumatic brain injury (TBI) and stroke [139]. When administered five minutes after injury, erythropoietin effectively reduced IL-1β, IL-6, and CXCL2 [140]. A one-hour dosing of erythropoietin prevented increased IL-1β and microglia later after injury in a model combining weight drop and hypoxia [141]. In a study involving 200 closed head injury patients with a Glasgow Coma Score > 3 [142], erythropoietin was compared with high or low hemoglobin transfusion. Transfusion, initiated within 6 h post-injury, aimed to maintain a hemoglobin threshold of 7 or 10 g/dL and included either erythropoietin or a placebo. In a separate observational study, erythropoietin therapy administered within the first 2 weeks post-injury resulted in patients on erythropoietin having significantly shorter stays in the intensive care unit, which is potentially suggestive of a longer survival [143]. The evidence for the use of erythropoietin has not reached the threshold for its use in a phase III trial [114].

Anakinra, a recombinant human IL-1 receptor antagonist (IL-1ra), was studied in a phase II randomized control clinical trial assessing neuroinflammatory modulation using anakinra following TBI [144]. This trial involved the study of 20 TBI patients with a Glasgow Coma Score of ≤ 8, who were recruited within the first 24 h after the injury. Using microdialysis probes within the brain parenchyma, various cytokines, including IL-1ra, were examined. CCL22 levels were reported to be significantly lowered in the anakinra group. The study was too small to establish anakinra as an effective clinical regimen but provided an intriguing approach for the use of extracellular fluid as a probe as opposed to baseline serum or CSF markers.

A limited number of previous studies have explored the use of statins for TBI, with two notable large observational trials. In one of these trials, conducted by Schneider et al., 523 patients with moderate to severe TBI (Abbreviated Injury Score of ≥3) were observed. Among the patients, 22% were regular users of statins [145]. The statin users were found to have a lower risk of in-hospital death. At a one-year assessment of the Extended Glasgow Outcome Scale of the 264 remaining patients, statin users had a small but significantly higher likelihood of more optimal recovery, but the net therapeutic effect of statins was not measurable once controlled for cardiovascular comorbidities in statin users.

### 4.8. Immunotherapy

Immunotherapy employing monoclonal antibodies has also been a subject of investigation in preclinical studies focused on tauopathies [84]. A recent study demonstrated that the delivery of an adeno-associated virus (AAV) vector coding for an anti-p tau antibody reduced CNS p tau levels in rodent models of repeated traumatic brain injury [146]. Furthermore, in an in vitro study, several tau antibodies demonstrated their efficacy in preventing neuronal tau uptake. Specifically, the antibody 6C5 successfully thwarted interneuronal propagation and the progression of tau aggregation after cellular uptake [147].

Specific antibodies targeting the pathogenic cis-P-tau post-TBI have been reported to lead to improved structural and functional outcomes [148,149,150]. Removing microglia with PLX5622, a colony-stimulating factor 1 receptor (CSF-1R), was found to have minimal impact on traumatic brain injury (TBI) outcomes. However, encouraging the turnover of these cells through pharmacological or genetic methods leads to a neuroprotective microglial phenotype and significant recovery after TBI. The positive effects of these replenished microglia rely heavily on interleukin-6 (IL-6) trans-signaling through the soluble IL-6 receptor (IL-6R) and strongly support adult neurogenesis [151].

### 4.9. Potential Dietary Targets

The consumption of a Western diet (WD) and the associated obesity have been consistently linked to systemic inflammatory responses, cognitive decline, and worsened outcomes following brain injuries [152,153,154]. WD-induced secretion of interleukins such as IL-1 b and IL-6 can disrupt neural circuits involved in cognition and memory [155]. In a preclinical study focused on the secondary injury outcomes resulting from a closed head injury (mTBI), obese C57 BL/6 mice fed a WD were compared to lean mice. At a chronic time point (30 days), the obese mice displayed significantly increased microglial activation and a chronic state of inflammation [156].

The Ketogenic diet (KD) is a fat-rich diet low in proteins and carbohydrates, with low obesogenic, and has been demonstrated to be neuroprotective [152,157,158]. Unlike the WD, the KD can reduce neuronal inflammation [159], rescue behavioral patterns of depression in animal models [160], diminish cognitive defects [161], and modulate neuronal injury [162]. In addition, Mediterranean diet (MD) consumption has also been associated with reduced risk of dementia and better memory and language performance [163]. Preclinical studies have reported that diets rich in anti-oxidants and flavonoids derived from fruits and vegetables effectively mitigate neuro-inflammation by modifying oxidative stress and apoptosis. This is achieved through the inhibition of NF-KB-dependent inflammatory signaling pathways [164].

## 5. Future Directions

Chronic traumatic encephalopathy (CTE) and traumatic encephalopathy syndrome (TES) have gained special attention in the public discourse. The diagnosis of CTE remains predominantly pathological, in turn, making the diagnosis of TES and emerging CTE pathological diagnosis challenging. As TES is a relatively novel clinical diagnostic classification, the exact prevalence of TES amongst athletes, combat veterans, and civilians remains largely unknown. A recent study encompassed 176 participants, consisting of 110 boxers and 66 mixed martial artists (MMA), who were all included in the analysis. Among them, 72 individuals (41% of the total) were categorized as having traumatic encephalopathy syndrome (TES), with the likelihood of TES increasing as age advanced. TES-positive (TES+) participants were more likely to be boxers, initiated their fighting careers at a younger age, engaged in more professional fights, and experienced more frequent knockouts [165].

There are a number of diagnostic challenges, which include limited research pertaining to diagnostic validity [164]. There are also limitations pertaining to the absence of universally agreed-upon biomarkers [20]. Despite proposed neuroimaging correlates for TBI and CTE, their diagnostic and prognostic utility remain elusive. Diagnostic challenges include small sample size, inherent heterogeneity of TBI/CTE among injured individuals, as well as the widely varying interval between injury and clinical assessment [84]. The majority of the completed studies lack female study participants, a significant limitation that hinders clinical applicability. Given the tauopathy nature of CTE/TES, wider use of tau markers, including tau PET ligands such as flortaucapir is needed [166]. Identification of sensitive and specific CTE/TES biomarkers would facilitate early diagnosis, monitoring of disease progression, and assessment of disease prognosis. Access to validated biomarkers would also provide the necessary basis to study the natural progression of the disease in a more systematic way.

Currently, there are no FDA-approved drugs for CTE/TES that would offer a disease-modifying effect. Although a number of preclinical studies have proposed potential therapeutic effects for CTE [95], there is an urgent need for large-scale clinical trials. Moreover, there are a number of proposed preclinical models of CTE [90,95,167,168,169], but there is no agreement on a specific animal model of CTE. The lissencephalic nature of the rodent cortex also adds a layer of complexity to its application to clinical studies. Moreover, the central role of neuroinflammation is increasingly recognized in TBI [84,170]. The development of disease-specific immunomodulating agents, including humanized monoclonal antibodies, is, quite possibly, on the CTE/TES treatment horizon [171].

Given the lack of current treatment options, the most viable CTE/TES treatment option is prevention and safe practices. There is a great need to continue the optimization of protective sports gear, vigilant enforcement of sports contact rules and protocols, and raising public awareness [172,173].

One area poised for significant future advancements is the development of highly sensitive and specific assays for traumatic brain injury (TBI) in serum and cerebrospinal fluid (CSF). Despite numerous studies, there is currently a lack of consensus regarding proposed biomarkers for TBI. The recent FDA approval of GFAP and UCHL-1 for acute assessment of TBI in the ED setting and examining the necessity of CT neuroimaging, is a major step forward [56,57]. Research on other TBI biomarkers has produced mixed results. For example, two prior studies found no correlation between serum S100B concentration and clinical outcomes, as measured by tools such as the Glasgow Outcome Scale (GOS), Glasgow Outcome Scale-Extended (GOS-E), or imaging studies [174,175]. Similarly, another study found that GFAP was not an effective clinical predictive marker based on GOS-E and functional timeline [176], while an earlier study demonstrated that serum-cleaved tau (c-tau) was also not a reliable predictor after mild TBI [177].

The research and clinical management of chronic traumatic encephalopathy (CTE) and traumatic encephalopathy syndrome (TES) are rapidly evolving areas as our current understanding of their neuropathological mechanisms continues to expand [10]. Advancements in ultra-sensitive biofluid assays are crucial for earlier and more accurate clinical detection of CTE/TES, as well as the potential for more effective treatments. In addition, the development and refinement of disease-specific markers such as tau PET ligands would further expand the much-needed arsenal for timely and effective management of this complex disease, both in research and clinical settings (Table 2).

## Figures and Tables

**Figure 1 biomedicines-11-03158-f001:**
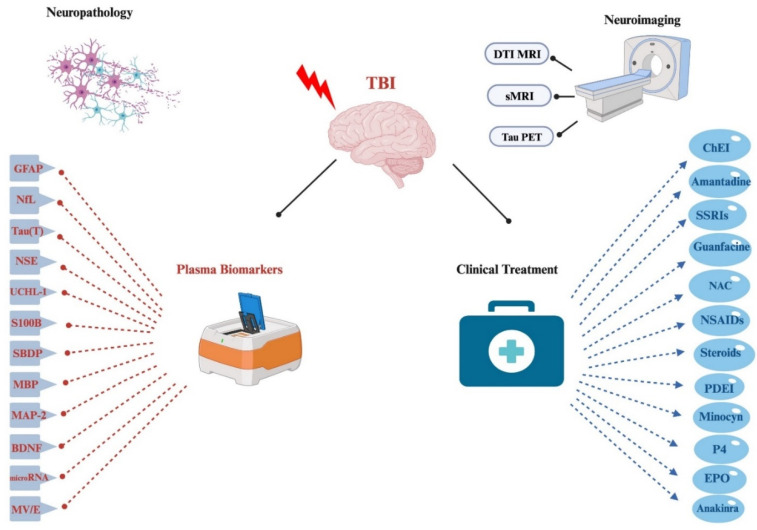
Schematic illustration of major plasma markers of TBI including GFAP (Glial Fibrillary Acidic Protein), NfL (Neurofibrillary Light Chain), total tau, NSE (Neuron Specific Enolase), UCHL-1 (Ubiquitin C-terminal hydrolase-1), S100B, SBDP (Spectrin Breakdown Products), MBP (Myelin Basic Protein), MAP-2 (Microtubule-Associated Protein-2), BDNF (Brain-Derived Neurotrophic Factor), microRNA, and MV/E (Microvesicles and Exosomes). Also included are the major clinical treatment options used in treatment of TBI patients, including cholinesterase inhibitors (ChEI), NMDA receptor antagonist (Amantadine), SSRIs, guanfacine, NSAIDs, Nutraceuticals such as NAC, phosphodiesterase inhibitor (PDEI), Minocycline (Minocyn), glucocorticoids and progesterone (P4), Erythropoietin (EPO) and Anakinra. Some of the current promising neuroimaging tools for clinical TBI patients include diffusion tensor imaging (DTI) imaging, structural MRI with corresponding volumetric analysis, as well as Tau PET imaging. Images created with BioRender.com (accessed on 22 October 2023).

**Figure 2 biomedicines-11-03158-f002:**
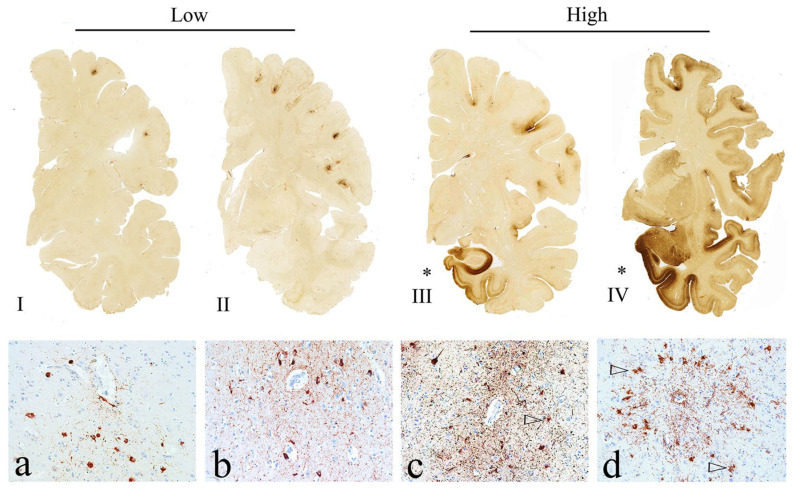
Top panel: Depiction of McKee staging system (**I**–**IV**). McKee stage I CTE is defined by one or two isolated CTE lesions at the depths of the cortical sulci. In stage II, there are typically three or more cortical CTE lesions. In stage III CTE, there are multiple loci of CTE lesions and diffuse NFTs in the medial temporal lobe. In stage IV CTE, CTE lesions and NFTs are ubiquitously distributed throughout the cerebral cortex, diencephalon, and brainstem. Bottom panel: Characteristic stages: (**a**) CTE stage I perivascular AT8 positive NFTs and neurites. (**b**) CTE stage II lesions comprised of several AT8 positive NFTs and neurites; (**c**) Characteristic stage III CTE lesions, comprised of several perivascular AT8 positive NFTs and neurites; (**d**) A large accumulation of several AT8 positive NFTs and neurites (Images adopted from McKee et al. *Acta Neuropathologica* 2023 [17]) The triangular shapes refer to neurofibrillary tangles and neuritis which are AT8 stain positive. * refers to extensive degeneration of amygdala and entorhinal cortex.

**Figure 3 biomedicines-11-03158-f003:**
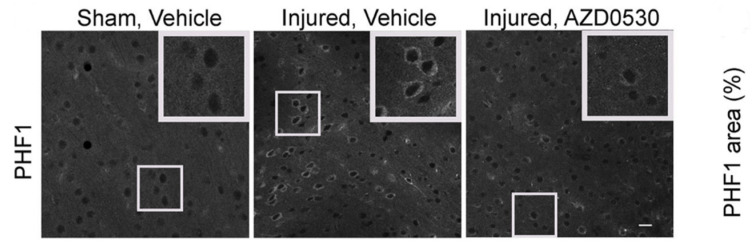
Representative images using immunofluorescent staining for PHF1 of coronal cerebral cortex sections within 0.5–1 mm medial to the site of injury in 7.5-month-old control mice from SV (Sham Vehicle treated), IV (Injured Vehicle treated), and IA (Injured AZD0530 treated) groups. (Images adopted from Tang et al. 2020 [90]).

**Table 1 biomedicines-11-03158-t001:** TBI Plasma Biomarkers. Summary of reported TBI serum markers including GFAP (Glial Fibrillary Acidic Protein), NfL (Neurofibrillary Light Chain), total tau, NSE (Neuron Specific Enolase), UCHL-1 (Ubiquitin C-terminal hydrolase-1), S100B, SBDP (Spectrin Breakdown Products), MBP (Myelin Basic Protein), MAP-2 (Microtubule-Associated Protein-2), BDNF (Brain-Derived Neurotrophic Factor), microRNA, and MV/E (Microvesicles and Exosomes).

Serum Biomarker	TBI Outcomes
GFAP	Clinical TBI studies have reported longitudinal elevation in GFAP levels [21]. GFAP was also recently approved by the FDA as a TBI outcome clinical measure [22].
NfL	Clinical TBI studies have reported elevated NfL serum levels both acutely and longitudinally [21].
Tau (total)	Total tau elevation has been reported both acutely and chronically in TBI populations [21].
NSE	NSE elevated levels have been reported in both mild and more severe TBI populations [23,24].
UCHL-1	UCHL-1 has been shown to be robustly elevated in both mTBI and more severe TBI patients [25]. UCHL-1 was recently FDA-approved as a TBI outcome clinical measure [22].
S100B	S100B has been reported to be more acutely elevated in various TBI severity cases [26,27].
SBDP	SBDPs are products of calpain and caspase-3 post-TBI and have been reported to be elevated in both preclinical and clinical studies [28,29].
MBP	MBP is an oligodendrocyte protein and a product of proteases, including calpain, and is reported to be elevated in severe TBI patients [30,31].
MAP-2	An emerging biomarker for TBI patients [32].
BDNF	Mainly reported in the preclinical TBI studies, with potential application to the clinical TBI population [33].
microRNA	A class of small endogenous RNA molecules that have been reported to be elevated in biofluid (CSF, serum, or plasma) in several rodent models of TBI of various severities [34].
MV/E	Lipid-bilayered, encapsulated particles (10–100 nm in diameter) that are released from cells into the CSF and blood during TBI [35]. Reported elevated MV/E released into CSF in TBI patients [36].
Pro-inflammatory cytokines (IL-6, IL-1, IL-8, IL-10, TNFα, CRP)	Pro-inflammatory markers, especially Il-6 and CRP, have been shown to have robust diagnostic and prognostic value [20].

**Table 2 biomedicines-11-03158-t002:** TBI pharmacological treatments reported in preclinical and clinical studies. An overview of the reported TBI pharmacological regimen examined in preclinical and clinical studies. A number of these regimens, including cholinesterase inhibitors, NMDA receptor antagonists, SSRIs, guanfacine, NSAIDs, as well as glucorticoids and progesterone, are cross-purposed medications that have been utilized in the treatment of TBI clinical studies. Tau phosphorylation, tau acetylation, and immunotherapy regimens have largely been examined in the preclinical TBI setting.

TBI Pharmacological Regimen	Proposed Mechanism
Cholinesterase Inhibitors	Cholinesterase inhibitors, including galantamine, donepezil, and rivastigmine, have been repurposed for TBI patients [84].
NMDA receptor antagonists	NMDA receptor antagonist, amantadine, has been shown to improve cognition in moderate to severe TBI patients [85].
SSRIs	Selective serotonin reuptake inhibitors (SSRIs) like sertraline and escitalopram have been utilized to mange behavioral symptoms in TBI patients [82].
Guanfacine	Guanfacine has been reported to improve working memory deficits in mild TBI patients [86].
Nutraceuticals	A number of nutraceuticals have been utilized in the treatment of TBI in preclinical and clinical studies, including N-acetylcysteine (NAC), flavonoids, resveratrol, alpha-tocopherol (vitamin E), coenzyme Q10 [87].
NSAIDs	COX-2 selective drugs like carprofen, celecoxib, meloxicam, nimesulide, and rofecoxib have undergone testing in various preclinical TBI models with no significant degree of established efficacy [114].
Glucocorticoids	Despite several promising preclinical studies, clinical trials have resulted in limited success, likely due to a narrow therapeutic window [127].
Phosphodiesterase Inhibitors	Phophodiesterase inhibitors have been utilized mostly in preclinical studies and have not been systematically studied in a clinical trial setting [133].
Minocycline	In prior preclinical studies, minocycline given between 5 min and 1 h after injury improved performance on a variety of neurobehavioral tests [136].
Progesterone	A large, multi-center Phase III PROtect III trial, as well as a second larger-scale trial (SYNAPSE), examined progesterone and did not establish clinical efficacy [178].
Erythropoietin	Despite preclinical studies’ success, the evidence for the use of erythropoietin has not reached the threshold for its use in a phase III trial [126].
Anakinra	A small phase II randomized controlled clinical trial reported anti-inflammatory benefits in an Anakinra-treated group; the study size was too small to establish efficacy but provided an intriguing potential future approach [144].
Tau phosphorylation targets	The studies focusing on tau-phosphorylation targets have been mostly preclinical, with possible future clinical applications [90].
Tau acetylation targets	Tau acetylation inhibitors, including salsalate and methylene blue, as well as histone deacetylase 6 and sirtuins, have largely been examined in the preclinical setting [104].
Immunotherapy	Specific antibodies targeting the pathogenic cis-P-tau post-TBI have been reported to lead to improved structural and functional outcomes [148] but have yet to be examined in larger clinical trial settings.

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
