# Peer review of "Navigating the Complexities of Traumatic Encephalopathy Syndrome (TES): Current State and Future Challenges"

_biomedicines, 2023, doi:10.3390/biomedicines11123158_

Round 1

Reviewer 1 Report

Comments and Suggestions for Authors

This is a very comprehensive review of TES. It encompasses all current understanding of the condition, including clinical trials and potential therapeutic approaches.

It is very likely that TES represent a heterogeneic mix of a spectrum of conditions, hence the difficulties in finding specific biomarkers and therapies.

in Table 1, include the reference for the pro inflammatory cytokines.

The manuscript is very well written and organized and easy to follow. I enjoyed reading this review.

Author Response

Response to Reviewer 1:

I am grateful for the comments of Reviewer 1.  This is much appreciated, as CTE/TES/TBI are the author’s primary area of research and clinical care, with a dedicated and vested interest.

I also completely agree with your comments regarding the heterogeneity of TES, and the likely challenges that would inevitably arise from studying TES and treating patients suffering from this condition.  I have also ensured that the corresponding references are provided in both tables 1 and 2.

Reviewer 2 Report

Comments and Suggestions for Authors

The Paper entitled “Navigating the Complexities of Traumatic Encephalopathy

Syndrome (TES): Current State and Future Challenges by Arman Fesharaki-Zadeh is an excellent contribution to the field of brain injury, both in clinical and pre-clinical research settings. According to the author, this review is a comprehensive compilation of the current understanding of the neuropathological and clinical features of Chronic traumatic encephalopathy, proposed biomarkers of traumatic brain injury in both research and clinical settings, and a range of treatments based on previous preclinical and clinical research studies.

Overall, the paper has been nicely presented with figures and simple illustrations.

The English style is perfect.

I will suggest adding tables, instead of long readings, the table may concisely summarize the findings.

For example, the author may add a table related to the current biomarkers.

Efforts have been made to present a huge amount of knowledge to the readers, but I will suggest shortening it and making it more precise.

Author Response

Response to Reviewer 2:

I greatly appreciate the comments of Reviewer 2 and I have made the following changes:

I have added the two tables, including the TES proposed biomarkers and treatment with the corresponding references.  Upon reviewing the manuscript, I also appreciated the length of the manuscript, and have decreased the quantity in the highlighted areas from 8245 words to 6685 words, with the efforts of making the manuscript more readable and accessible.